# Single-Base Gene Variants in *MIR-146A* and *SCN1A* Genes Related to the Epileptogenic Process in Drug-Responsive and Drug-Resistant Temporal Lobe Epilepsy—A Preliminary Study in a Brazilian Cohort Sample

**DOI:** 10.3390/ijms25116005

**Published:** 2024-05-30

**Authors:** Renata Parissi Buainain, André Rodrigues Sodré, Jéssica Silva dos Santos, Karen Antonia Girotto Takazaki, Luciano de Souza Queiroz, Carlos Tadeu Parisi de Oliveira, Paulo Henrique Pires de Aguiar, Fernando Augusto Lima Marson, Manoela Marques Ortega

**Affiliations:** 1Laboratory of Cell and Molecular Tumor Biology and Bioactive Compounds, São Francisco University, Bragança Paulista 12916-900, SP, Brazil or fernandolimamarson@hotmail.com (F.A.L.M.); 2Laboratory of Molecular Biology and Genetics, São Francisco University, Bragança Paulista 12916-900, SP, Brazil; 3Department of Pathology, Faculty of Medical Science, University of Campinas, Campinas 13083-970, SP, Brazil; gradanat@unicamp.br; 4São Francisco University Hospital, São Francisco University, Bragança Paulista 20210-030, SP, Brazil; 5Department of Neurosurgery, Hospital Santa Paula, São Paulo 04556-100, SP, Brazil

**Keywords:** drug-resistant, drug-responsive, *MIR-146a* gene, *SCN1A* gene, single nucleotide variant, temporal lobe epilepsy

## Abstract

The drug-resistant temporal lobe epilepsy (TLE) has recently been associated with single nucleotide variants (SNVs) in microRNA(miR)-146a (*MIR-146A*) (rs2910164) and Sodium Voltage-Gated Channel Alpha Subunit 1 (*SCN1A*) (rs2298771 and rs3812718) genes. Moreover, no studies have shown an association between these SNVs and susceptibility to drug-resistant and drug-responsive TLE in Brazil. Thus, deoxyribonucleic acid (DNA) samples from 120 patients with TLE (55 drug-responsive and 65 drug-resistant) were evaluated by real-time polymerase chain reaction (RT-PCR). A total of 1171 healthy blood donor individuals from the Online Archive of Brazilian Mutations (ABraOM, from Portuguese Arquivo Brasileiro On-line de Mutações), a repository containing genomic variants of the Brazilian population, were added as a control population for the studied SNVs. *MIR-146A* and *SCN1A* relative expression was performed by quantitative RT-PCR (qRT-PCR). The statistical analysis protocol was performed using an alpha error of 0.05. TLE patient samples and ABraOM control samples were in Hardy–Weinberg equilibrium for all studied SNVs. For rs2910164, the frequencies of the homozygous genotype (CC) (15.00% vs. 9.65%) and C allele (37.80% vs. 29.97%) were superior in patients with TLE compared to controls with a higher risk for TLE disease [odds ratio (OR) = 1.89 (95% confidence interval (95%CI) = 1.06–3.37); OR = 1.38 (95%CI = 1.04–1.82), respectively]. Drug-responsive patients also presented higher frequencies of the CC genotype [21.81% vs. 9.65%; OR = 2.58 (95%CI = 1.25–5.30)] and C allele [39.09% vs. 29.97%; OR = 1.50 (95%CI = 1.01–2.22)] compared to controls. For rs2298771, the frequency of the heterozygous genotype (AG) (51.67% vs. 40.40%) was superior in patients with TLE compared to controls with a higher risk for TLE disease [OR = 2.42 (95%CI = 1.08–5.41)]. Drug-resistant patients presented a higher AG frequency [56.92% vs. 40.40%; OR = 3.36 (95%CI = 1.04–17.30)] compared to the control group. For rs3812718, the prevalence of genotypes and alleles were similar in both studied groups. The *MIR-146A* relative expression level was lower in drug-resistant compared to drug-responsive patients for GC (1.6 vs. 0.1, *p*-value = 0.049) and CC (1.8 vs. 0.6, *p*-value = 0.039). Also, the *SCN1A* relative expression levels in samples from TLE patients were significantly higher in AG [2.09 vs. 1.10, *p*-value = 0.038] and GG (3.19 vs. 1.10, *p*-value < 0.001) compared to the AA genotype. In conclusion, the rs2910164-CC and rs2298771-AG genotypes are exerting significant risk influence, respectively, on responsive disease and resistant disease, probably due to an upregulated nuclear factor kappa B (NF-kB) and *SCN1A* loss of function.

## 1. Introduction

Epilepsy, a chronic disease of the central nervous system, affects individuals of all ages [1] with an average world incidence of around 5.4 per 1000 individuals and an average world lifetime incidence of approximately 7.0 per 1000 individuals [1,2]. The World Health Organization estimates that 50 million people are diagnosed with epilepsy in the world, with 4 million in Brazil [3,4]. Still, about 66% of the epilepsy cases are temporal lobe epilepsy (TLE), classified as focal epilepsy [5]. It is the most common epileptic syndrome in adults, of which 40% are drug-resistant seizures [6].

More than half of epilepsy cases present a genetic basis and a complex inheritance pattern [7,8]. The single nucleotide variants (SNVs) alter amino acids of protein-coding genes and can influence protein function and play a vital role in the pathophysiology of diseases such as epilepsy [9]. SNVs at microRNA (miR)-146a (*MIR-146A*) and voltage-gated sodium channel *SCN1A* (Sodium Voltage-Gated Channel Alpha Subunit 1) have been recently related to TLE [10,11,12].

MiRs are small non-coding molecules that bind to messenger RNA (mRNA) and prevent its translation, and recent studies have observed miRs related to epilepsy [12,13]. miR-146a is upregulated in human astrocytes in epileptogenic tissues and it regulates the inflammatory process through nuclear factor-kappa B (NF-kB) signaling [14,15]. The SNV n.60G>C (rs2910164) located at the mature sequence of miR-146a has been studied in Italian, Chinese, and Brazilian populations with contradictory results [11,16,17,18]. In addition, only one Chinese and one Brazilian study have evaluated patients with TLE [11,18].

Epileptic seizure susceptibility versus multidrug resistance has recently been related to the different genotypes from SNVs in the *SCN1A* gene, including SNV c.3184A>G (rs2298771) and SNV IVS5N+5G>A (rs3812718) [19,20,21,22,23,24,25,26,27,28]. There is only one study that has evaluated the SNV rs3812718 and risk in relation to non-multidrug resistance in patients with TLE [29].

Drug-resistant epilepsy is defined as a failure of adequate trials of two tolerated, appropriately chosen, and used antiepileptic drug schedules (whether as monotherapies or in combination) to achieve sustained seizure freedom [30]. Thus, the main goal of the preliminary study was to show for the first time the association of SNV rs2910164 in the *MIR-146A* gene, SNVs rs2298771 and rs3812718 in the *SCN1A* gene, and the susceptibility to drug-resistant and drug-responsive TLE in a Brazilian cohort sample.

## 2. Results

### 2.1. Patients with Temporal Lobe Epilepsy and Healthy Controls

This study’s groups comprised 120 patients with TLE, 55 patients diagnosed as drug-responsive (22 males; 33 females; mean age: 45.23 years), and 65 patients diagnosed as drug-resistant (27 males; 38 females; mean age: 51.03 years) (Table 1). The average seizure onset age was earlier in the drug-resistant than in the drug-responsive group (11.09 vs. 23.81 years; *p*-value = 0.004). The average of Caucasian patients was higher than 70% in both TLE groups (*p*-value = 0.001). The sides of epileptiform paroxysms were similar in both TLE groups (*p*-value = 0.066) (Table 1). However, it was observed that brain tissue injury was more significant in the drug-resistant than in the drug-responsive group (90.80% vs. 40.70%; *p*-value < 0.001) (Table 1). The drug-resistant group used anti-epileptic drug polytherapy in 96.90% of the cases, while the drug-responsive group used monotherapy in 72.20% of the cases (*p*-value < 0.001) (Table 1).

### 2.2. Distribuion of the Genotype and Allele Frequencies for Single Nucleotide Variants (SNVs) rs2910164, rs2298771, and rs3812718 Using Online Archive of Brazilian Mutations (ABraOM, from Portuguese Arquivo Brasileiro On-Line de Mutações) Controls

TLE patient samples and Online Archive of Brazilian Mutations (ABraOM, from Portuguese Arquivo Brasileiro On-line de Mutações) control samples were in Hardy–Weinberg equilibrium for all studied SNVs (Table 2, Table 3 and Table 4).

For rs2910164, the frequencies of the homozygous genotype (CC) (15.00% vs. 9.65%, *p*-value = 0.043) and C allele (37.80% vs. 29.97%) were superior in patients with TLE compared to controls with a higher risk for the disease [Odds Ratio (OR) = 1.89 (95% confidence interval (95%CI) = 1.06–3.37); OR = 1.38 (95%CI = 1.04–1.82), respectively] (Table 2). Drug-responsive patients also presented higher frequencies of the CC genotype [(21.81% vs. 9.65%; OR = 2.58 (95%CI = 1.25–5.30)] and C allele [(39.09% vs. 29.97%; OR = 1.50 (95%CI = 1.01–2.22)] compared to controls (Table 2).

For rs2298771, the frequency of the heterozygous genotype (AG) (51.67% vs. 40.40%) was superior in patients with TLE compared to controls with a higher risk for the disease [OR = 2.42 (95%CI = 1.08–5.41)] (Table 3). In addition, drug-resistant patients also presented a higher AG genotype frequency [(56.92% vs. 40.40%; OR = 3.36 (95%CI = 1.04–17.30)] compared to the control group. However, for drug-responsive patients, the genotypes and prevalence of alleles were similar in patients and controls (Table 3). For rs3812718, the genotypes and prevalence of alleles were similar in both studied groups (Table 4).

In addition, no differences occurred in the association between the groups of patients, namely the drug-resistant and drug-responsive groups.

### 2.3. microRNA (miR)-146a (MIR-146A) and Sodium Voltage-Gated Channel Alpha Subunit 1 (SCN1A) Quantification Considering Single Nucleotide Variants (SNVs) rs2910164 and rs2298771

The *MIR-146A* relative expression levels in samples of TLE patients were lower in GC (1.02 vs. 1.63, *p*-value = 0.234] and CC (1.03 vs. 1.63, *p*-value = 0.491) genotypes compared to the GG genotype (Figure 1A); however, no significant association was observed. Moreover, when patients were divided into drug-resistant and drug-responsive groups, the *MIR-146A* relative expression level was lower in drug-resistant compared to drug-responsive patients for GG (1.3 vs. 2.0, *p*-value = 0.275), and it was only significantly lower for GC (1.6 vs. 0.1, *p*-value = 0.049) and CC (1.8 vs. 0.6, *p*-value = 0.039) genotypes (Figure 1B), possibly indicating an increased NF-kB inflammation process in drug-resistant patients harboring GC or CC genotypes for SNV rs2910164.

In contrast, the *SCN1A* relative expression levels in samples from TLE patients were significantly higher in the AG genotype [2.09 vs. 1.10, *p*-value = 0.038] and GG (3.19 vs. 1.10, *p*-value < 0.001) compared to the AA genotype (Figure 1C). The results may indicate an *SCN1A* loss of function in patients with TLE harboring AA genotypes for SNV rs2298771. In addition, drug-responsive patients presented no significant *SCN1A* expression levels for AA (1.1 vs. 1.1; *p*-value = 0.894), AG (2.6 vs. 1.7; *p*-value = 0.107), and GG (2.6 vs. 3.8, *p*-value = 0.237) genotypes compared with drug-resistant patients (Figure 1D).

The *MIR-146A* and *SCN1A* relative expression values for each evaluated sample are presented in Appendix A.

## 3. Discussion

In the present preliminary report, we performed a case–control study to analyze two potentially functional SNVs of the *MIR-146A* and *SCN1A* genes and the risk of epilepsy in a Brazilian cohort sample. Previous studies have suggested that decreased miR-146a expression may be associated with increased NF-kB inflammation and susceptibility to the development of epilepsy [11,12]. Moreover, *MIR-146A* was observed with an increased expression level in human brain astrocytes, and it may inhibit target genes related to epileptic inflammatory process [31].

Only three previous studies have evaluated the SNV rs2910164 and TLE. The first study observed that the rs2910164 variant in the pre-miR146a gene is unlikely to influence the risk of developing TLE or its severity in Italian patients [16]. In the second study, Cui et al. (2015) concluded that the rs2910164 variant was not associated with TLE [18]. The third study from our group suggested that the GC genotype for SNV rs2910164 appears to be associated with susceptibility to drug-resistant TLE in Brazilian patients, probably due to the decreased *MIR-146A* expression, favoring the NF-kB pathway [11]. The CC genotype for SNV rs2910164 could also be related to susceptibility to drug-resistant TLE, but there was a small number of patients harboring the CC genotype. Thus, to confirm the results, we evaluated a higher number of drug-resistant TLE patients. The same cohort of drug-resistant patients evaluated previously was included in the present study. Additionally, more than 15 drug-resistant patients were also added to the present study. In Boschiero et al. (2020), only three drug-resistant patients were identified as harboring the CC genotype for SNV rs2910164 [11]; here, only three drug-resistant patients were identified as the CC genotype and no risk for TLE was observed, indicating that a larger Brazilian cohort should be evaluated for SNV rs2910164. The hospital at Bragança Paulista (São Paulo, Brazil) is not known for its high complexity, and it is not prepared to receive many difficult-to-manage patients. Moreover, the Brazilian population is mixed, with their origins mainly being from Europeans, Amerindians, Africans, Levantines, and East Asians, explaining our contrasting results [32,33]. Thus, the discrepancy in our results might be due to the ethnic variation and differences in the number of recruited patients.

Corroborating with our previous publication [11], the *MIR-146A* expression level was lower in drug-resistant compared to drug-responsive patients for GC and CC genotypes, indicating an increased NF-kB inflammation process in drug-resistant patients harboring the GC or CC genotypes. Interestingly, when drug-responsive TLE patients were evaluated, the CC genotype was related to the disease susceptibility compared to control individuals with a 2.6-fold risk for drug-responsive patients with TLE. In fact, the *MIR-146A* expression level was lower in drug-responsive patients harboring the CC variant, indicating increased NF-kB.

The α subunit of the voltage-gated sodium channel is a large protein of 2000 amino acids and its function is to generate a brief influx of sodium ions by transiently opening in response to neuronal membrane depolarization and closing within milliseconds [34]. Single amino acid substitutions can alter numerous components of channel function and deviate from normal channel function, causing clinical consequences such as epilepsy [35]. Thus, our study evaluated the SNVs rs2298771 and rs3812718, two of the most common SNPs in the intron and exon of the *SCN1A* gene, influencing the regulation of the gene expression and its structure and functionality, respectively; also, both SNVs are closely related to resistance to sodium-channel-blocking antiepileptic drugs [19].

Two studies have found a correlation between the A allele and combined genotypes GA + AA for SNV rs2298771 and the poor response to antiepileptic sodium channel blockers [19,22]. In contrast, five studies have found no association between genotypes from SNV rs2298771 and sodium channel blocker metabolism or resistance [20,21,23,24,25].

To our knowledge, our study is the first to evaluate the SNV rs2298771 in TLE patients. Interestingly, we found that both the AA genotype and A allele present an increased risk for the disease in drug-responsive patients. In fact, the *SCN1A* expression level was lower in drug-responsive patients harboring the AA genotype. Our results seem to indicate that the wild-type genotype for SNV rs2298771 in *SCN1A*, which encodes the voltage-gated sodium channel—NaV1.1 sodium channel alpha subunit—results in a loss-of-function protein presenting a lower expression, with an insufficiency of NaV1.1, as observed in the *SCN1A* gene mutated in Dravet Syndrome, as well as milder phenotypes associated with genetic epilepsy with febrile seizures plus [34,36,37].

Seven studies have evaluated the susceptibility for general epilepsy seizure, multidrug resistance, and the SNV rs3812718 [19,20,22,23,26,27,28]. Only one study has analyzed TLE risk and the SNV rs3812718 [23]. Thus, the authors compared genotypes and allele frequencies between South Indian Ancestry patients with mesial TLE with hippocampal sclerosis (mTLE-HS) and observed that the AA genotype and A allele were overrepresented in these patients, contributing to increased susceptibility to mTLE-HS [23]. Only one study has demonstrated an association between the genotype harboring the A allele for the SNV rs3812718 and the need to administer higher doses of anti-epileptic drugs than those with the GG genotype, whereas a correlation with the multidrug resistance phenotype was not detectable [26]. Our present study demonstrated no susceptibility to TLE, drug-responsive, or drug-resistant patients, for the SNV rs3812718.

Generally, the cohort of patients with seizure onset during childhood exhibit a worse response to medication. This underscores the need for a deeper understanding of epilepsy as a phenotype reflecting developmental-aging processes characterized by maladaptive neuroplasticity. These processes begin before conception and continue through pregnancy, childhood, and subsequent critical periods such as adolescence and reproductive senescence, posing risks across the lifespan [38].

Over the past decade, significant progress has been made in understanding the genetic and morphogenic mechanisms underlying cortical malformations and developmental brain tumors. Focal malformations are primarily caused by somatic variants in genes associated with cell growth, particularly in the mechanistic Target of Rapamycin (mTOR) pathway for focal cortical dysplasia type 2, acquired in neuronal progenitors during neurodevelopment. Conversely, developmental brain tumors arise from somatic variants in genes linked to cell proliferation, such as the Mitogen-activated protein (MAP)-kinase pathway in ganglioglioma, affecting proliferating glioneuronal precursors. The timing and specific gene involved during neurodevelopment determine the nature and size of the lesion, whether it manifests as a developmental malformation or a brain tumor [39].

SNPs are genetic variations that can influence susceptibility to diseases, including epilepsy. These SNPs may impact gene regulation, protein function, and other biological processes relevant to the development and pathogenesis of epilepsy. Therefore, SNPs could potentially play a role in determining individual susceptibility to epilepsy, including drug-resistant forms, and may contribute to the complex genetic interactions underlying this condition.

Thus, in our drug-responsive cohort, the CC variant genotype of SNP rs2910164 in the *MIR-146A* gene is more prevalent and patients exhibit lower *MIR-146A* expression, particularly in genotypes GC and CC, suggesting increased NF-kB-mediated inflammation in drug-responsive cases harboring the CC genotype. For SNP rs2298771 in the *SCN1A* gene, drug-resistant patients more frequently have the AG genotype versus AA + GG. Higher expression levels of *SCN1A* are observed in AG and GG genotypes, indicating that the wild-type genotype may lead to a loss-of-function NaV1.1 protein in drug-resistant cases harboring AG or GG genotypes. Our results need to be confirmed in a larger drug-responsive and drug-resistant cohort. In conclusion, the rs2910164-CC and/or rs2298771-AG genotypes are exerting significant risk influence, respectively, on responsive disease and resistant disease, probably due to an upregulated NF-kB and *SCN1A* loss of function.

## 4. Materials and Methods

### 4.1. Research Ethics Committee

This study was approved by the Ethics Committee of São Francisco University (approval no 45723615.0.0000.5514).

### 4.2. Temporal Lobe Epilepsy Patient Selection and Control Population

The selection of patients with epilepsy was performed from an electronic medical record system at the hospital. Thus, a total of 70 patients with TLE were enrolled, with 55 being drug-responsive and 15 being drug-resistant. For the TLE diagnosis and seizure classification, we evaluated the personal and family history of epilepsy, clinical and neurological physical examination, electroencephalogram and/or video-electroencephalogram, magnetic resonance imaging, or computed tomography. In addition, this study comprised 50 samples of human drug-resistant TLE tissues obtained from surgical amygdalohippocampectomy patients between January 2015 and December 2018. All epilepsy paraffin-embedded tissues were donated from Prof. Dr. Luciano de Souza Queiroz, Department of Pathology, University of Campinas, São Paulo, Brazil. The patients’ inclusion criteria were based on the International League Against Epilepsy [30,40,41,42].

The control population for the association study comprised 1171 healthy blood donor individuals from the ABraOM database, a repository containing genomic variants of the Brazilian population, with a total of 77,236,632 variants (September 2020) [43].

### 4.3. Deoxyribonucleic Acid (DNA) Samples and Single Nucleotide Variants (SNVs) Identification

Ten milliliters of peripheral venous blood was collected from 55 drug-responsive and 15 drug-resistant patients. Further, genomic deoxyribonucleic acid (DNA) samples for genotyping were isolated using lithium chloride extraction [44]. Genomic DNAs were previously isolated from 50 drug-resistant patient tissues [11] using a phenol- and chloroform-based protocol [45].

The *MIR-146A* (rs2910164) and *SCN1A* (rs2298771 and rs3812718) genotypes were identified using real-time polymerase chain reaction (RT-PCR) performed on the StepOne RT-PCR (Applied Biosystems^®^, Waltham, MA, USA) using the standard TaqMan^®^ genotyping assay (rs2910164: C_15946974_10; rs2298771: C_11748767_20; rs3812718: C_25982233_10) according to the manufacturer’s instructions.

### 4.4. Quantitative Real-Time Polymerase Chain Reaction

Total ribonucleic acid (RNA) was isolated using Trizol^®^ Reagent (Invitrogen™, Carlsbad, CA, USA) from peripheral blood or paraffin-embedded epileptic tissue samples [rs2910164: GG n = 12; GC n = 10; CC n = 7; rs2298771: TT n = 10; CT n = 9; CC n = 6] according to the manufacturer’s instructions. *MIR-146A* (assay 000468) and U6 (assay 001973) complementary DNA (cDNA) were synthesized from total RNA according to the TaqMan^®^ real-time assays protocol (Applied Biosystems^®^). The relative expression of each target was quantified by the delta–delta cycle threshold (ΔΔCt) method [46]. Each sample was examined in triplicate and the raw data are presented as the relative quantity of the target, normalized by U6. For *SCN1A* analyses, cDNA conversion from total RNA was performed using a High-Capacity cDNA Reverse Transcription Kit (Applied Biosystems^®^, USA). Each sample was examined in triplicate and the expression of each gene was normalized by the control gene glyceraldehyde-3-phosphate dehydrogenase (*GAPDH*) and calculated by applying the 2−ΔΔCt method. The *MIR-146A* and *SCN1A* expression means for each genotype for the SNVs rs2910164 and rs2298771 (wild-type; heterozygous; variant) were evaluated by T-Test. Primer sequences used for amplification by quantitative real-time polymerase chain reaction (qRT-PCR) with the SYBRGreen dye (Applied Biosystems^®^, USA) are as follows: *SCN1A* (forward) 5′-AGGCTGGAATATCTTTGACGG-3′ and (reverse) 5′-GCCAACTTGAAAACTCGCAG-3′; *GAPDH* (forward) 5′-CCACTTGATTTTGGAGGGAT-3′ and (reverse) 5′-GCACCGTCAAGGCTGAGAAC-3′.

### 4.5. Statistical Analyses

The Hardy–Weinberg equilibrium was tested using the Chi-square test. Differences between groups were analyzed using the Chi-square test or Fisher Exact test for categorical data (genotype and allele frequencies). In addition, the comparison between groups for numerical data was performed using the T-test (gene expression) and Mann–Whitney test (patients’ age and age of onset). The normality of the numerical data was evaluated using the Shapiro–Wilk test and the Kolmogorov–Smirnov test. For all statistical tests, significance is two-sided and achieved when *p*-values are less than 0.05. Moreover, a correction for multiple comparisons was performed using the Bonferroni approach. The corrected *p*-values are presented only in the tables and figures legends. All tests were performed using the Statistical Package for the Social Sciences (IBM Corp. Released 2012. IBM SPSS Statistics for Windows, Version 21.0. Armonk, NY, USA: IBM Corp.). The figure was drawn using the GraphPad Prism version 10.0.0 for Mac, GraphPad Software, Boston, MA, USA, www.graphpad.com (accessed on 16 March 2024).

## 5. Conclusions

For SNV rs2910164 (*MIR-146A*), a significantly increased frequency of variant genotype (CC) was observed in drug-responsive patients with TLE. The *MIR-146A* relative expression level was lower in drug-resistant compared to drug-responsive patients for GC and CC genotypes versus the GG genotype for SNV rs2910164, indicating an increased NF-kB inflammation process in drug-resistant patients harboring GC or CC genotypes.

For SNV rs2298771 in the *SCN1A* gene, an increased frequency of the AG genotype versus AA + GG genotypes was observed in drug-resistant patients with TLE. The *SCN1A* relative expression level was higher in AG and GG genotypes versus the AA genotype for SNV rs2298771, indicating that the wild-type genotype of *SCN1A*, which encodes the NaV1.1 sodium channel alpha subunit, might result in the *SCN1A* loss-of-function protein presenting a lower expression with an NaV1.1 insufficiency.

## Figures and Tables

**Figure 1 ijms-25-06005-f001:**
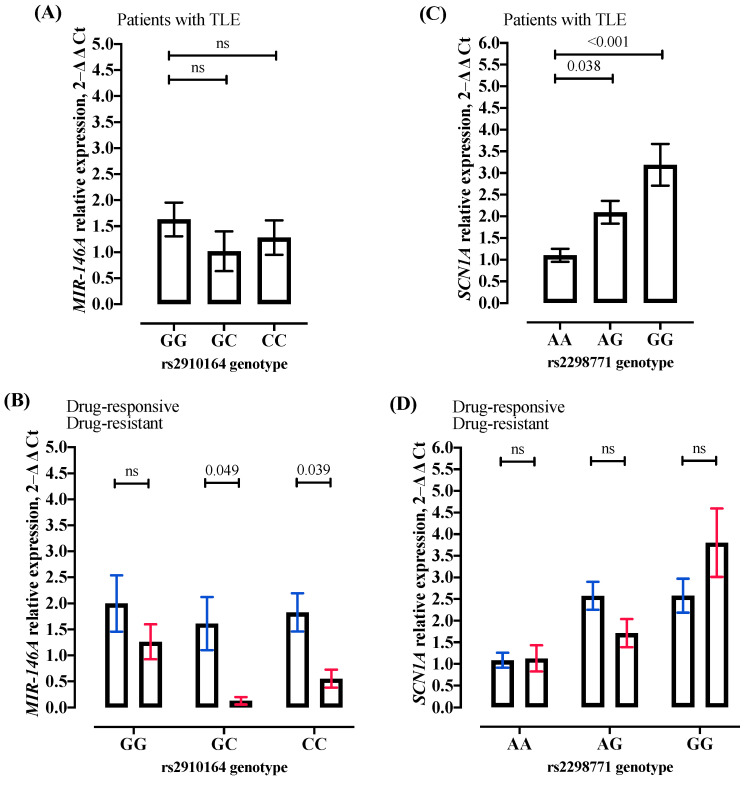
Quantitative real-time polymerase chain reaction (qRT-PCR) for microRNA (miR)-146a *(MIR-146A)* and Sodium Voltage-Gated Channel Alpha Subunit 1 (*SCN1A*) gene expression. (**A**) *MIR-146A* gene expression from patients with temporal lobe epilepsy (TLE) for each genotype for the single nucleotide variant (SNV) rs2910164 [GG (n = 12); GC (n = 10); CC (n = 7)]. (**B**) *MIR-146A* gene expression from patients with drug-responsive and drug-resistant TLE for each genotype for SNV rs2910164 [drug-responsive: GG (n = 6); GC (n = 6); CC (n = 4); drug-resistant: GG (n = 6); GC (n = 4); CC (n = 3]. (**C**) *SCN1A* gene expression from patients with TLE for each genotype for SNV rs2298771 [AA (n = 10); AG (n = 9); GG (n = 6)]. (**D**) *SCN1A* gene expression from patients with drug-responsive and drug-resistant TLE for each genotype for SNV rs2298771 [drug-responsive: AA (n = 6); AG (n = 4); GG (n = 3); drug-resistant: AA (n = 4); AG (n = 5); GG (n = 3]. All values are represented as mean ± standard deviation. The gene expression was evaluated in patient epileptogenic tissues or peripheral blood from patients with TLE. The statistical analysis was performed using T-Test. Also, all *p*-values lower than 0.042 were significant after the correction for multiple tests using the Bonferroni approach. 2−∆∆Ct algorithm, the delta–delta cycle threshold; ns, no significant; n, number of individuals (samples).

**Table 1 ijms-25-06005-t001:** Clinical variables of the patients with temporal lobe epilepsy enrolled in this study.

Markers	Groups	Patients		*p*-Value (*p*-Corrected)
		Drug-Resistant n (%)	Drug-Responsive n (%)	
Sex	Female	38 (58.50)	33 (60.00)	1.000 (1.000) ^a^
	Male	27 (41.50)	22 (40.00)	
Age (years) *		53.50 (49.51–56.00)	45.00 (33.00–39.00)	0.118 (0.236) ^c^
Race	White people	51 (78.50)	39 (70.90)	0.001 (0.002) ^b^
	*Pardos* (Mixed race)	13 (20.00)	4 (7.30)	
	Black people	1 (1.50)	10 (18.20)	
	Asian individuals	0 (0.00)	2 (3.60)	
Age of onset (years) *		10.50 (7.00–13.00)	15.00 (12.00–19.00)	0.004 (0.008) ^c^
Electroencephalogram	Not specified	2 (3.10)	5 (9.10)	0.066 (0.132) ^b^
	Normal	0 (0.00)	5 (9.10)	
	Bilateral temporal	12 (18.50)	8 (14.50)	
	Right temporal	22 (33.80)	18 (32.70)	
	Left temporal	29 (44.60)	19 (34.50)	
Structural brain lessions	Yes	59 (90.80)	22 (40.70)	<0.001 (<0.001) ^a^
	No	6 (9.20)	32 (59.30)	
Therapy with antiepileptic drug	Monotherapy	2 (3.10)	39 (72.20)	<0.001 (<0.001) ^b^
	Polytherapy	63 (96.90)	15 (27.80)	

^a^, Chi-square test; ^b^, Fisher’s exact test; ^c^, Mann–Whitney test; *, median [95% confidence interval (95%CI)]; %, percentage; n, number of individuals. *p*-corrected was adjusted using Bonferroni correction for multiple comparisons.

**Table 2 ijms-25-06005-t002:** Comparative association of the variant rs2910164 in microRNA (miR)-146a (*MIR-146A*) gene and susceptibility to temporal lobe epilepsy and Online Archive of Brazilian Mutations (ABraOM, from Portuguese Arquivo Brasileiro On-line de Mutações) controls.

Genotypes and Alleles	Patients n (%) ^a^	Controls n (%) ^b^	*p*-Value (*p*-Corrected)	OR (95%CI)
GG	49 (40.83)	582 (49.70)		Reference
GC	53 (44.17)	476 (40.65)	0.213 * (1.000)	1.32 (0.88–1.99)
**CC**	**18 (15.00)**	**113 (9.65)**	**0.043 * (1.000)**	**1.89 (1.06–3.37)**
GG + CC	67 (55.83)	695 (59.35)		Reference
GC	53 (44.17)	476 (40.65)	0.517 * (1.000)	1.16 (0.79–1.69)
GG + GC	102 (85.00)	1058 (90.35)		Reference
CC	18 (15.00)	113 (9.65)	0.090 * (1.000)	1.65 (0.37–2.83)
GC + CC	71 (59.17)	589 (50.30)		Reference
GG	49 (40.83)	582 (49.70)	0.079 * (1.000)	0.70 (0.48–1.02)
**Allele C**	**89 (37.08)**	**702 (29.97)**	**0.028 * (0.672)**	**1.38 (1.04–1.82)**
Allele G	151 (62.92)	1640 (70.03)		Reference
**Genotypes and Alleles**	**Drug-Resistant n (%) ^c^**	**Controls n (%)**	***p*-Value (*p*-Corrected)**	**OR (95%CI)**
GG	25 (38.46)	582 (49.70)		Reference
GC	34 (52.31)	476 (40.65)	0.078 * (1.000)	1.66 (0.98–2.83)
CC	6 (9.23)	113 (9.65)	0.836 * (1.000)	1.24 (0.50–3.08)
GG + CC	31 (47.69)	695 (59.35)		Reference
GC	34 (52.31)	476 (40.65)	0.084 * (1.000)	1.60 (0.97–2.64)
GG + GC	59 (90.77)	1058 (90.35)		Reference
CC	6 (9.23)	113 (9.65)	0.917 * (1.000)	0.95 (0.40–2.25)
GC + CC	40 (61.54)	589 (50.30)		Reference
GG	25 (38.46)	582 (49.70)	0.102 * (1.000)	0.63 (0.38–1.06)
Allele C	46 (35.38)	702 (29.97)	0.227 * (1.000)	1.28 (0.88–1.85)
Allele G	84 (64.62)	1640 (70.03)		Reference
**Genotypes and Alleles**	**Drug-Responsive n (%) ^d^**	**Controls n (%)**	***p*-Value (*p*-Corrected)**	**OR (95%CI)**
GG	24 (43.63)	582 (49.70)		Reference
GC	19 (34.55)	476 (40.65)	0.958 * (1.000)	0.97 (0.52–1.79)
**CC**	**12 (21.82)**	**113 (9.65)**	**0.015 * (0.360)**	**2.58 (1.25–5.30)**
GG + CC	36 (65.45)	695 (59.35)		Reference
GC	19 (34.55)	476 (40.65)	0.447 * (1.000)	0.77 (0.44–1.36)
GG + GC	43 (78.19)	1058 (90.35)		Reference
**CC**	**12 (21.81)**	**113 (9.65)**	**0.007 * (0.168)**	**2.61 (1.34–5.10)**
GC + CC	31 (56.36)	589 (50.30)		Reference
GG	24 (43.64)	582 (49.70)	0.459 * (1.000)	0.78 (0.45–1.35)
**Allele C**	**43 (39.09)**	**702 (29.97)**	**0.050 * (1.000)**	**1.50 (1.01–2.22)**
Allele G	67 (60.91)	1640 (70.03)		Reference
**Genotypes and Alleles**	**Drug-Resistant n (%) ^c^**	**Drug-Responsive n (%) ^d^**	***p*-Value (*p*-Corrected)**	**OR (95%CI)**
GG	25 (38.46)	24 (43.63)		Reference
GC	34 (52.31)	19 (34.55)	0.255 * (1.000)	1.72 (0.78–3.80)
CC	6 (9.23)	12 (21.82)	0.314 * (1.000)	0.480 (0.16–1.48)
GG + CC	31 (47.69)	36 (65.45)		Reference
GC	34 (52.31)	19 (34.55)	0.077 * (1.000)	2.08 (0.99–4.35)
GG + GC	59 (90.77)	43 (78.19)		Reference
CC	6 (9.23)	12 (21.81)	0.095 * (1.000)	0.36 (0.13–1.05)
GC + CC	40 (61.54)	31 (56.36)		Reference
GG	25 (38.46)	24 (43.64)	0.698 * (1.000)	0.81 (0.39–1.68)
Allele C	46 (35.38)	43 (39.09)	0.647 * (1.000)	0.85 (0.51–1.44)
Allele G	84 (64.62)	67 (60.91)		Reference

*, Chi-square with Yates correction; %, percentage; 95%CI, 95% confidence interval; n, number of individuals; OR, odds ratio. *p*-value for Hardy–Weinberg equilibrium: (a) 0.842; (b) 0.555; (c) 0.510; (d) 0.126. Values below 0.05 indicate that the sample is out of Hardy–Weinberg equilibrium. Bold type indicates the presence of statistically significant differences in the proportions of genotypes or alleles between the groups of patients and controls. An alpha error of 0.05 was adopted in the statistical analysis. *p*-corrected was adjusted using Bonferroni correction for multiple comparisons.

**Table 3 ijms-25-06005-t003:** Comparative association of the variant rs2298771 in Sodium Voltage-Gated Channel Alpha Subunit 1 (*SCN1A*) gene and susceptibility to temporal lobe epilepsy and Online Archive of Brazilian Mutations (ABraOM, from Portuguese Arquivo Brasileiro On-line de Mutações) controls.

Genotypes and Alleles	Patients n (%) ^a^	Controls n (%) ^b^	*p*-Value (*p*-Corrected)	OR (95%CI)
AA	51 (42.50)	569 (48.60)	0.298 * (1.000)	1.65 (0.73–3.72)
**AG**	**62 (51.67)**	**473 (40.40)**	**0.027 * (0.648)**	**2.42 (1.08–5.41)**
GG	7 (5.83)	129 (11.00)		Reference
AA + GG	58 (48.33)	698 (59.60)		Reference
**AG**	**62 (51.67)**	**473 (40.40)**	**0.022 * (0.528)**	**1.58 (1.08–2.30)**
AA + AG	113 (94.17)	1042 (89.00)		Reference
GG	7 (5.83)	129 (11.00)	0.109 * (1.000)	0.50 (0.23–1.10)
AG + GG	69 (57.50)	602 (51.40)		Reference
AA	51 (42.50)	569 (48.60)	0.240 * (1.000)	0.78 (0.54–1.14)
Allele G	76 (31.70)	731 (68.79)		Reference
Allele A	164 (68.30)	1611 (31.31)	0.943 * (1.000)	0.98 (0.74–1.30)
**Genotypes and Alleles**	**Drug-Resistant n (%) ^c^**	**Controls n (%)**	***p*-Value (*p*-Corrected)**	**OR (95%CI)**
AA	25 (38.46)	569 (48.60)	0.438 ** (1.000)	1.89 (0.56–9.92)
**AG**	**37 (56.92)**	**473 (40.40)**	**0.041 ** (0.984)**	**3.36 (1.04–17.30)**
GG	3 (4.62)	129 (11.00)		Reference
AA + GG	28 (43.08)	698 (59.60)		Reference
**AG**	**37 (56.92)**	**473 (40.40)**	**0.012 * (0.288)**	**1.95 (1.18–3.23)**
AA + AG	62 (95.38)	1042 (89.00)		Reference
GG	3 (4.62)	129 (11.00)	0.136 ** (1.000)	0.39 (0.08–1.23)
AG + GG	40 (61.54)	602 (51.40)		Reference
AA	25 (38.46)	569 (48.60)	0.143 * (1.000)	0.66 (0.40–1.10)
Allele G	43 (33.10)	731 (68.79)		Reference
Allele A	87 (66.90)	1611 (31.31)	0.727 * (1.000)	0.92 (0.63–1.34)
**Genotypes and Alleles**	**Drug-Responsive n (%) ^d^**	**Controls n (%)**	***p*-Value (*p*-Corrected)**	**OR (95%CI)**
AA	26 (47.27)	569 (48.60)	0.665 ** (1.000)	1.47 (0.50–5.91)
AG	25 (45.46)	473 (40.40)	0.464 ** (1.000)	1.70 (0.57–6.86)
GG	4 (7.27)	129 (11.00)		Reference
AA + GG	30 (54.55)	698 (59.60)		Reference
AG	25 (45.45)	473 (40.40)	0.455 * (1.000)	1.23 (0.71–2.12)
AA + AG	51 (92.72)	1042 (89.00)		Reference
GG	4 (7.28)	129 (11.00)	0.537 ** (1.000)	0.63 (0.16–1.77)
AG + GG	29 (52.72)	602 (51.40)		Reference
AA	26 (47.28)	569 (48.60)	0.958 * (1.000)	0.95 (0.55–1.63)
Allele G	33 (33.00)	731 (68.79)		Reference
Allele A	77 (77.00)	1611 (31.31)	0.871 * (1.000)	1.06 (0.70–1.61)
**Genotypes and Alleles**	**Drug-Resistant n (%) ^c^**	**Drug-Responsive n (%) ^d^**	***p*-Value (*p*-Corrected)**	**OR (95%CI)**
AA	25 (38.46)	26 (47.27)	1.000 ** (1.000)	1.28 (0.19–9.61)
AG	37 (56.92)	25 (45.46)	0.645 ** (1.000)	1.95 (0.30–14.50)
GG	3 (4.62)	4 (7.27)		Reference
AA + GG	28 (43.08)	30 (54.55)		Reference
AG	37 (56.92)	25 (45.45)	0.286 * (1.000)	1.59 (0.77–3.27)
AA + AG	62 (95.38)	51 (92.72)		Reference
GG	3 (4.62)	4 (7.28)	0.814 ** (1.000)	0.62 (0.09–3.84)
AG + GG	40 (61.54)	29 (52.72)		Reference
AA	25 (38.46)	26 (47.28)	0.431 * (1.000)	0.70 (0.34–1.44)
Allele G	43 (33.10)	33 (33.00)		Reference
Allele A	87 (66.90)	77 (77.00)	0.710 * (1.000)	0.87 (0.50–1.50)

*, Chi-square with Yates correction; **, Fisher’s exact test; %, percentage; 95%CI, 95% confidence interval; n, number of individuals; OR, odds ratio. *p*-value for Hardy–Weinberg equilibrium: (a) 0.105; (b) 0.127; (c) 0.070; (d) 0.830. Values below 0.05 indicate that the sample is out of Hardy–Weinberg equilibrium. Bold type indicates the presence of statistically significant differences in the proportions of genotypes or alleles between the groups of patients and controls. An alpha error of 0.05 was adopted in the statistical analysis. *p*-corrected was adjusted using Bonferroni correction for multiple comparisons.

**Table 4 ijms-25-06005-t004:** Comparative association of the variant rs3812718 in Sodium Voltage-Gated Channel Alpha Subunit 1 (*SCN1A*) gene and susceptibility to temporal lobe epilepsy and Online Archive of Brazilian Mutations (ABraOM, from Portuguese Arquivo Brasileiro On-line de Mutações) controls.

Genotypes and Alleles	Patients n (%) ^a^	Controls n (%) ^b^	*p*-Value (*p*-Corrected)	OR (95%CI)
GG	30 (25.00)	245 (20.90)	0.144 * (1.000)	1.54 (0.90–2.62)
GA	60 (50.00)	549 (46.90)	0.211 * (1.000)	1.37 (0.87–2.17)
AA	30 (25.00)	377 (32.20)		Reference
GG + AA	60 (50.00)	622 (53.10)		Reference
GA	60 (50.00)	549 (46.90)	0.579 * (1.000)	1.13 (0.78–1.65)
GG + GA	90 (75.00)	794 (67.80)		Reference
AA	30 (25.00)	377 (32.20)	0.131 * (1.000)	0.70 (0.46–1.08)
GA + AA	90 (75.00)	926 (79.00)		Reference
GG	30 (25.00)	245 (21.00)	0.357 * (1.000)	1.26 (0.81–1.95)
Allele A	120 (50.00)	1303 (50.66)		Reference
Allele G	120 (50.00)	1269 (49.34)	0.898 * (1.000)	1.03 (0.79–1.34)
**Genotypes and Alleles**	**Drug-Resistant n (%) ^c^**	**Controls n (%)**	***p*-Value (*p*-Corrected)**	**OR (95%CI)**
GG	17 (26.15)	245 (20.90)	0.090 * (1.000)	2.01 (0.96–4.22)
GA	35 (53.85)	549 (46.90)	0.084 * (1.000)	1.85 (0.97–3.54)
AA	13 (20.00)	377 (32.20)		Reference
GG + AA	30 (46.15)	622 (53.10)		Reference
GA	35 (53.85)	549 (46.90)	0.334 * (1.000)	1.32 (0.80–2.18)
GG + GA	52 (80.00)	794 (67.80)		Reference
AA	13 (20.00)	377 (32.20)	0.055 * (1.000)	0.53 (0.28–0.98)
GA + AA	48 (73.85)	926 (79.00)		Reference
GG	17 (26.15)	245 (21.00)	0.396 * (1.000)	1.34 (0.76–2.37)
Allele A	61 (46.90)	1303 (50.66)		Reference
Allele G	69 (53.10)	1269 (49.34)	0.458 * (1.000)	1.16 (0.82–1.65)
**Genotypes and Alleles**	**Drug-Responsive n (%) ^d^**	**Controls n (%)**	***p*-Value (*p*-Corrected)**	**OR (95%CI)**
GG	13 (23.64)	245 (20.90)	0.810 * (1.000)	1.18 (0.56–2.47)
GA	25 (45.45)	549 (46.90)	0.897 * (1.000)	1.01 (0.54–1.90)
AA	17 (30.91)	377 (32.20)		Reference
GG +AA	30 (54.55)	622 (53.10)		Reference
GA	25 (45.45)	549 (46.90)	0.945 * (1.000)	0.94 (0.55–1.63)
GG + GA	38 (69.09)	794 (67.80)		Reference
AA	17 (30.91)	377 (32.20)	0.959 * (1.000)	0.94 (0.53–1.69)
GA + AA	42 (76.36)	926 (79.00)		Reference
GG	13 (23.64)	245 (21.00)	0.754 * (1.000)	1.17 (0.62–2.21)
Allele A	59 (53.60)	1303 (50.66)		Reference
Allele G	51 (46.40)	1269 (49.34)	0.607 * (1.000)	0.89 (0.60–1.30)
**Genotypes and Alleles**	**Drug-Resistant n (%) ^c^**	**Drug-Responsive n (%) ^d^**	***p*-Value (*p*-Corrected)**	**OR (95%CI)**
GG	17 (26.15)	13 (23.64)	0.439 * (1.000)	1.71 (0.62–4.76)
GA	35 (53.85)	25 (45.45)	0.264 * (1.000)	1.83 (0.76–4.44)
AA	13 (20.00)	17 (30.91)		Reference
GG + AA	30 (46.15)	30 (54.55)		Reference
GA	35 (53.85)	25 (45.45)	0.464 * (1.000)	1.40 (0.68–2.89)
GG + GA	52 (80.00)	38 (69.09)		Reference
AA	13 (20.00)	17 (30.91)	0.245 * (1.000)	0.56 (0.24–1.29)
GA + AA	48 (73.85)	42 (76.36)		Reference
GG	17 (26.15)	13 (23.64)	0.916 * (1.000)	1.14 (0.41–2.63)
Allele A	61 (46.90)	59 (53.60)		Reference
Allele G	69 (53.10)	51 (46.40)	0.365 * (1.000)	1.31 (0.79–2.18)

*, Chi-square with Yates correction; %, percentage; 95%CI, 95% confidence interval; n, number of individuals; OR, odds ratio: (a) 1.000; (b) 0.228; (c) 0.808; (d) 0.816. Values below 0.05 indicate that the sample is out of Hardy–Weinberg equilibrium. An alpha error of 0.05 was adopted in the statistical analysis. *p*-corrected was adjusted using Bonferroni correction for multiple comparisons.

## Data Availability

The original contributions presented in the study are included in the article/Appendix A, further inquiries can be directed to the corresponding authors.

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
