# Peer review of "Single-Base Gene Variants in MIR-146A and SCN1A Genes Related to the Epileptogenic Process in Drug-Responsive and Drug-Resistant Temporal Lobe Epilepsy—A Preliminary Study in a Brazilian Cohort Sample"

_ijms, 2024, doi:10.3390/ijms25116005_

Round 1

Reviewer 1 Report

Comments and Suggestions for Authors

The authors demonstrated that the association between SNVs and drug resistance in patients with TLE. However, there were several concerns about this study. First, there was no definitions for drug resistant epilepsy. Some patients were taking only one antiseizure medication despite of drug-resistant epilepsy group. Were they really a drug-resistant epilpesy group? Second, In the method section, 15 patients had drug-resistant epilepsy, but the numbers were different in the result secton. Correction is needed. Third, Because multiple genes were compared, multiple correction should be necessary. In addition, since the groups are compared between drug-resistant group, drug-responsive group, and healthy control group, multiple correction is naturally necessary.

Reviewer 2 Report

Comments and Suggestions for Authors

The authors present interesting findings supporting associations between single-base gene variants in MIR-146a and SCN1A genes related to epileptogenesis resulting in drug resistant or responsive epilepsy in an adult population compared to case-controlled healthy subjects who did not express epilepsy.

These important findings require greater time-dependent context regarding their study group's life-course history beginning with reproductive, prenatal and childhood contributions that contributed to the adverse outcome of epilepsy. The authors should include greater details regarding reproductive, pregnancy and childhood risks or disease exposures, or include a bridging limitation section in the discussion that states a plan to explore these  details with further investigations. This will require more detailed explanations in their text supported by peer-reviewed literature that is not presently in their manuscript.

Worse response to medication was reported for the subset in their cohort with seizure onset during childhood. This is a provocative finding that requires greater explanation for the reader to appreciate epilepsy as a phenotype that represents developmental-aging processes involving maladaptive neuroplasticity that cumulatively begins before conception, continue  during pregnancy and are followed by childhood diseases or adverse events, starting during the first 1000 days. Subsequent critical/sensitive time periods include adolescence and following reproductive senescence to reflect risks across the lifespan (Scher Frontiers in Neurology doi: 10.3389/fneuro.2023.1321674).

Epilepsy is an adverse outcome that represents a continuum of developmental-aging toxic stressor interplay involving multiple disease pathways. Gene-environment interactions adversely affect the neural exposome associated with multiple endogenous-exogenous communicable and noncommunicable events/factors with epilepsy as an abnormal outcome (Scher Seminars in Ped Neurol 2022).

Current research regarding epileptogenesis supports parental or de novo mitotic processes at conception that can be clinically activated by post-translational abnormal development of progenitor neuronal populations( microglial, microglial, and interneuron subtypes), resulting in an impaired neuronal connectome. This can occur  particularly in more susceptible brain regions such as the temporal lobe regions.

Neuropathological examination using advance neuroimmunological markers have identified specific components within the neuron involving the developing synapses that are more often documented in drug-resistant children and adults requiring surgical resections for drug-resistant focal epilepsies associated with focal dysplasia or brain tumors (Blumcke et al Lance Neurol Nov 2021, based on there earlier publication in Epilepssia that same year.

This Brazilian cohort also includes subsets that represent various degrees of health disparities. Epilepsy is an chronic condition that requires the study of diversity, equity and inclusion to mitigate these diaprties in more vulnerable populations.

Round 2

Reviewer 1 Report

Comments and Suggestions for Authors

Thank you for making the effort to appropriately revise the manuscript with my previous suggestion. However, despite these efforts, it is difficult for this paper to be accepted due to the limitations mentioned previously.

Author Response

Reply was send directely to the Editor.

Reviewer 2 Report

Comments and Suggestions for Authors

appreciate the revisions

Author Response

Reply was sent directly to the Editor.